# Male circumcision uptake during the Botswana Combination Prevention Project

**Tafireyi Marukutira**[1], **Faith Ussery**[2], **Etienne Kadima**[3], **Lisa A. Mills**[1], **Jan Moore**[2], **Lisa Block**[2,4], **Pam Bachanas**[2], **Stephanie Davis**[2], **Tracey Schissler**[5], **Roselyn Mosha**[5], **Onneile Komotere**[5], **Thebeyame Diswai**[5], **Conrad Ntsuape**[6], **Refeletswe Lebelonyane**[6], **Naomi Bock**[2]*

1 Centers for Disease Control and Prevention, Gaborone, Botswana, 2 Division of Global HIV/AIDS, Centers for Disease Control and Prevention, Atlanta, Georgia, United States of America, 3 Botswana-Harvard AIDS Institute Partnership, Gaborone, Botswana, 4 Northrop Grumman Corporation, Atlanta, Georgia, United States of America, 5 Jhpiego, Gaborone, Botswana, 6 Department of HIV/AIDS Prevention and Care, Ministry of Health, Gaborone, Botswana

* nnbock731@gmail.com

## Abstract

### Introduction

Voluntary medical male circumcision (VMMC) uptake has been slow in some countries, including Botswana. To inform demand creation efforts, we examined sociodemographic characteristics and referral procedures associated with VMMC uptake in the Botswana Combination Prevention Project (BCPP) and examined the effectiveness of referral of men to MC services from HIV testing venues.

### Design

BCPP was a community-randomized trial evaluating the impact of a combination HIV prevention package which included VMMC on community HIV incidence. We conducted a sub-analysis of VMMC uptake in intervention communities.

### Methods

During the initial VMMC campaign in 15 intervention communities, baseline male circumcision (MC) status was assessed among men eligible for HIV testing. Uncircumcised male community residents aged 16–49 years with negative/unknown HIV status were mobilized and linked to study VMMC services. Outcomes included MC baseline status and uptake through study services. Univariate and multivariate logistic regressions were performed to identify factors associated with MC uptake.

### Results

Of 12,864 men eligible for testing, 50% (n = 6,448) were already circumcised. Among the uncircumcised men (n = 6,416), 10% (n = 635) underwent MC. Of the 5,071 men identified as eligible for MC through HIV testing services, 78% declined referral and less than 1% of those were circumcised. Of those accepting referral (n = 1,107), 16% were circumcised.

**Data Availability Statement:** The data relevant to this study are third party data. The data is available at the CDC from https://data.cdc.gov/Global-Health/Botswana-Combination-Prevention-Project-BCPP-

Publi/qcw5-4m9q. The authors confirm they did not have special access privileges that others would not have.

**Funding:** This project has been supported by the President's Emergency Plan for AIDS Relief (PEPFAR) through the Centers for Disease Control and Prevention (CDC) under the terms of Cooperative Agreements U2G GH000073 and U2G GH000419. The funders, the U.S. CDC, designed the study, collected the data, and led the data analysis.

**Competing interests:** The authors have declared that no competing interests exist.

Younger (16–24 years) (aOR: 1.51; 95%CI:1.22,1.85), unemployed men (aOR:1.34; 95% CI: 1.06,1.69), and those undergoing HIV testing at mobile venues (aOR: 1.88; 95%CI: 1.53,2.31) were more likely to get circumcised. Fear of pain was the most prevalent (27%) reason given for not being circumcised.

## Conclusion

Younger, unemployed men seeking HIV testing at mobile sites in Botswana were more likely to get VMMC. Addressing unique barriers for employed and older men may be necessary. Given the simplicity of VMMC as an intervention, the HIV testing programs offer a platform for identifying uncircumcised men and offering information and encouragement to access services.

## Introduction

HIV prevention remains key to efforts to end the AIDS epidemic by 2030. Male circumcision (MC) is one of the five Joint United Nations Program on HIV and AIDS (UNAIDS) pillars of HIV prevention [1, 2]. MC has been shown to reduce HIV acquisition by men from heterosexual sex by approximately 60%, making voluntary male medical circumcision (VMMC) programs a crucial component of the HIV prevention portfolio in countries with generalized epidemics in Eastern and Southern Africa [3–5]. MC has also been shown to be cost-effective, as a one-time intervention which provides lifelong partial protection against female-to-male HIV transmission [6].

Currently, the US President's Plan for Emergency AIDS Relief (PEPFAR) and other donors support free VMMC services for men in 15 sub-Saharan African countries [7]. VMMC has been scaled up gradually in the past decade, with more than 26 million circumcisions performed between 2008 and 2019 in the priority countries. However, this uptake has been unevenly distributed across countries, with some lagging, including Botswana. In total, 241 539 medical circumcisions were performed in Botswana between program launch in 2008 and 2020 [8], peaking in 2013 and stagnating afterwards while other countries accelerated year over year. Botswana thus fell short of initial national and global coverage targets (80% of HIV negative 15–49 years old males by 2016), with an estimated VMMC coverage of 43% in 2016 year [7–10]. Few tribal groups still offer circumcisions at initiation ceremonies in Botswana; hence, there was no other major source of circumcision beyond the struggling VMMC program [11, 12].

VMMC is culturally acceptable and there is high patient satisfaction in Botswana even though there is need for continuous involvement of influential community leaders [13–15]. As spontaneous public demand has not been sufficient to achieve VMMC coverage targets despite these facts, programs have come to rely on outreach activities designed to motivate potential clients to seek VMMC, termed demand creation activities. These commonly include interpersonal mobilization (trained mobilizers in direct conversation with potential clients), mass media (educational and promotional spots on radio and television), and targeted service delivery (services offered in convenient times, places and ways for their target populations) [16]. However, an additional pathway is recommended by the World Health Organization (WHO) but has not been evaluated: use of HIV testing services (HTS) to identify eligible men, inform them about MC, and refer them to services when desired [17]. Universal HIV testing could

offer a comprehensive method for identifying men who may be eligible for MC services. Given that men frequently do not access health care services unless they are ill, community-based HIV testing may provide access to a population of men who would not otherwise seek health care. As with all demand creation methods, the place of this pathway within a portfolio of demand creation interventions would depend on both its overall yield and the characteristics of the men it tends to attract.

The Botswana Combination Prevention Project (BCPP) provided an opportunity to test this pathway as a new approach to addressing Botswana's chronically limited uptake of VMMC. BCPP implemented multiple community-based interventions to reduce HIV incidence in its intervention communities, including a VMMC component with referral of men from HIV testing to MC services [18]. Here, we examine the uptake of VMMC among uncircumcised men through this pathway in intervention communities, and the demographic characteristics predicting obtaining circumcision in this way, as well as those associated with circumcision at baseline. To our knowledge, though barriers, facilitators, attitudes and outcomes around MC have been studied in Botswana, this is one of the first published evaluations of specific interventions to improve uptake in that setting.

## Methodology

### Study design

The BCPP was a pair-matched community-randomized HIV prevention trial designed to evaluate the impact of a package of combination prevention (CP) services on HIV incidence in Botswana. A full description of the study design is available elsewhere and registered on clinicaltrials.gov (NCT01965470) [18]. The study was conducted in 30 rural or peri-urban communities across the southern, central and northern regions of Botswana. The 30 communities were selected based on desired size and accessibility. Matched-pair randomization was used to ensure that control and intervention groups were balanced on community size, baseline access to health services including ART, age structure, and geographic location. The matched community pairs were randomized 1:1 to the intervention group or the control group. The analysis reported in this paper includes data from the 15 intervention communities, as data on VMMC were only collected in those communities. The data here are from the first round of interventions conducted between October 2013 and February 2016. Men were given information about MC availability in the community through two different phases of the study: 1) during a BCPP baseline survey of a random sample of 20% of households and 2) during the BCPP HIV testing campaign conducted in the home and at various outreach sites throughout the communities that included referral to MC services and follow-up. The testing and MC campaigns were conducted for a 6–8 week period in each community to ensure that the remaining 80% of the community households were enumerated and offered HIV testing.

### Participants

HIV-negative men and those with unknown HIV status aged 16–49 years who were citizens or spouses of citizens resident in the intervention communities were eligible for the MC referral. Men were assessed for circumcision at baseline through self-report. Although VMMC services were offered to all uncircumcised HIV-negative men or men with unknown HIV status, only citizens or spouses of citizens who were residents in the intervention communities are included in these analyses.

## Procedures

Eligible men were asked if they had been circumcised. Circumcision status was based on self-report with no distinction made between medical and nonmedical self-reported circumcision. Men identified as uncircumcised through the BCPP community HIV testing campaign were told "*We are offering free safe male circumcision services as part of BCPP, including help with transportation. Would you like to be referred to the circumcision tent to learn more? If you decide to, you can also receive circumcision there.*" After agreeing, the man was assigned an appointment date and included on an electronic encrypted referral list provided to the VMMC team on a daily basis. The VMMC team then used this list to conduct phone-based follow-up. First, up to three telephone attempts to reach each man directly were made; the first within a week of referral and each subsequent one occurring weekly. Upon reaching the man, mobilizers provided education and encouragement around circumcision and addressed questions until he either indicated disinterest or scheduled an MC appointment.

## Outcomes

Circumcision status was determined through self-report at baseline. Circumcision uptake was objectively documented following the procedure as part of the intervention, in a purpose-built study database, by the study staff serving as the circumcision providers.

## Data collection

Interviewer-administered questionnaires collected self-reported demographics, sexual risk behavior, and HIV testing data on encrypted handheld tablets and data were stored in the main database. VMMC referral activities including dates and outcomes of contact attempts and circumcision date were captured by study VMMC staff in a separate database built with Epi-info™ v7.1.5 [19]. To better understand the uptake of MC among these men, we examined potential failures in service delivery. The unique national identifier (Omang) was collected to identify and link individuals across these databases.

## Data analysis

Descriptive analysis using proportions and means was used to compare sociodemographic factors by circumcision status at baseline and by circumcision up-take. To account for clustering of observations within communities, the Rao-Scott Chi-square test was used to test for these associations [20]. We evaluated the association between circumcision at baseline and sociodemographic factors as well as the association between circumcision up-take and sociodemographic factors in unadjusted and adjusted logistic regression models accounting for clustering. SAS 9.4 was used for all statistical analyses. PROC SURVEYLOGISTIC in SAS was used to account for clustering at the community-level [21].

## Ethical review

The study was approved by the Centers for Disease Control and Prevention (CDC) Institutional Review Board (Protocol #6475) and the Botswana Health Research and Development Committee (HRDC); Institutional Review Board of the Botswana Ministry of Health and Wellness). The study was monitored by an Independent Data Safety Monitoring Board.

The VMMC procedure was conducted according to the standards for routine VMMC service delivery in Botswana and the WHO minimum package of services, including written informed consent, sexually transmitted infections screening and treatment, condom education

and provision, and postoperative follow-up for documenting healing [17]. Minors aged 16 or 17 years required parental/guardian written consent and client assent.

## Results

Of the 12,864 men identified through BCPP, 50% (n = 6,448) reported already being circumcised at baseline. Among the 6,416 men not circumcised at baseline, 10% (n = 635) underwent VMMC as part of study intervention during the 6–8 weeks of the HIV testing campaign.

### Sociodemographic characteristics of men by circumcision status at baseline

Characteristics of the men by their baseline circumcision status are in Table 1. There was no difference in age observed between uncircumcised men and circumcised men at baseline (*P* = 0.09). Among circumcised men, 16.5% were married compared to 13.7% among

**Table 1. Characteristics of men aged 16–49 years circumcised vs. not circumcised at baseline, N = 12,864.**

| Characteristic | Circumcised (N = 6,448) n (%) | Uncircumcised (N = 6,416) n (%) | *p*-value[*] |
|---|---|---|---|
| **Age** | | | 0.09 |
| 16–24 | 2,616 (40.6%) | 2,594 (40.4%) | |
| 25–34 | 2,251 (34.9%) | 2,403 (37.5%) | |
| 35–49 | 1,581 (24.5%) | 1,419 (22.1%) | |
| **Age (Mean, SD)** | 28.3 (8.6) | 28.1 (8.3) | |
| **Marital status[a]** | | | <0.001[**] |
| Single/never married | 5,355 (83.0%) | 5,505 (85.8%) | |
| Married | 1,063 (16.5%) | 881 (13.7%) | |
| Divorced/separated/widowed | 30 (<1%) | 29 (<1%) | |
| **Education[b]** | | | <0.001[**] |
| Primary/lower | 756 (11.7%) | 1,025 (16.0%) | |
| Secondary or higher | 5,691 (88.3%) | 5,388 (84.0%) | |
| **Employment[c]** | | | <0.001[**] |
| Unemployed | 3,296 (51.1%) | 3,562 (55.6%) | |
| Employed | 3,150 (48.9%) | 2,842 (44.4%) | |
| **Sex partners in past 12 months[d]** | | | < .0001[**] |
| 2 or more partners | 579 (9.0%) | 774 (12.1%) | |
| 1 partner | 4,175 (64.9%) | 3,998 (62.6%) | |
| No partners | 1,680 (26.1%) | 1,617 (25.3%) | |
| **Alcohol use in past 3 months[e]** | | | <0.001[**] |
| Yes | 2,580 (40.0%) | 2,870 (44.7%) | |
| No | 3,866 (60.0%) | 3,544 (55.3%) | |
| **Test Venue** | | | <0.001[**] |
| Mobile | 3,365 (52.2%) | 2,685 (41.8%) | |
| Home | 3,083 (47.8%) | 3,731 (58.2%) | |

*n*, number of participants; SD, standard deviation

[*]Rao-Scott Chi-Square test;

[**]Significant at α = .05

[a]1 participant missing marital status.

[b]3 participants missing education level.

[c]13 participants missing employment status.

[d]41 participants missing number of sex partners.

[e]4 participants missing alcohol usage.

uncircumcised men (*P*<0.001). A higher proportion of circumcised men had secondary or higher education (88.3% vs 84.0%, *P*<0.001) and employed (48.9% vs. 44.4%, *P*<0.001). Among circumcised men, 9.0% reported having at least 2 or more sex partners in the past 12 months compared to 12.1% of uncircumcised men (*P*<0.001). A higher proportion of circumcised men reported alcohol use (60% reporting none in the past 3 months, vs. 55.3% of uncircumcised men). The majority of uncircumcised men (58.2%) were found in the home whereas the majority of circumcised men (52.2%) were found through mobile testing (*P*<0.001). In a multivariate analysis younger age, more education, being employed, fewer sex partners in the past year, less alcohol use, and a higher likelihood of having HIV testing at mobile venues was associated with being circumcised at baseline.

At baseline, fear of pain was the most prevalent (27%) reason given for not being circumcised. Other reasons given for not being circumcised were not having time/money for VMMC (11%), not interested (10%) and never offered the procedure (9%).

### Participant characteristics by VMMC uptake among men uncircumcised at baseline

Among the 6,416 men not circumcised at baseline, 10% (n = 635) of the uncircumcised men underwent VMMC (Table 2). Among men who underwent VMMC, 50.1% were aged 16–24 years (*P*<0.001). The majority of men who underwent VMMC were unemployed (65.5%; *P* = 0.001) and were referred from the mobile unit (54.5%; *P*<0.001). Men who underwent VMMC reported at least 1 sex partner (52.7%; *P* <0.01) and being single (92.0%; *P*<0.001). There were no observed differences in education levels and alcohol use (*P* = 0.21, 0.43, respectively).

In multivariate analyses, men were more likely to become circumcised if aged 16–24 years compared to those aged 35–49 years (aOR: 1.51; 95%CI: 1.22,1.85; *P* = 0.001), with intake through mobile units compared to home testing (aOR: 1.88; 95%CI:1.53,2.31; *P*<0.001), and unemployed compared to employed (aOR: 1.34; 95%CI:1.06,1.69; *P* = 0.02). Men who were circumcised during the study did not have significantly higher odds of having 2 or more sex partners (aOR: 1.03; 95%CI: 0.65,1.64; *P* = 0.88) or being single (aOR: 1.35; 95%CI: 0.99,1.84; *P* = 0.06).

### Uptake of MC by men referred through HIV testing and counseling

Among the 6,416 uncircumcised men, the majority of men (n = 5,071) were first identified through BCPP community HIV testing and the remainder (n = 1,345) were first contacted by the study's VMMC mobilization team. Among the 5,071 men identified through community HIV testing, 213 (4.2%) were circumcised through study services. The majority (n = 3,964) of 5,071 (78%) refused referral to MC services. The primary reasons given for refusal included not being ready to make a decision (35%), fear of pain (21%), and planning MC at a future date (15%). Of the refusers, 35 men (less than one percent) were circumcised during the study. Of the 1,107 men who agreed to referral, 178 (16%) were circumcised by BCPP services.

### Discussion

BCPP found half of HIV-negative or HIV status unknown men to be circumcised at baseline and successfully circumcised an additional 10% during a 6–8 week intervention period per community. The 50% circumcision coverage at baseline was unexpected and higher than reported in the 2013 Botswana AIDS Indicator Survey (BAIS) (24% among males 10–64 years, with similar coverage among older age groups) [9]. Through its national VMMC program, Botswana has conducted VMMC campaigns and provided routine services [22–24].

**Table 2. Demographics and risk factors of initially uncircumcised men who did vs. did not undergo VMMC during the study, N = 6,416.**

| Characteristic | Underwent VMMC (N = 635) n (%) | Did not undergo VMMC (N = 5,781) n (%) | p-value* | Unadjusted Odds Ratio (95% CI);p-value | Adjusted Odds Ratio[a] (95% CI);p-value |
|---|---|---|---|---|---|
| **Age** | | | <0.001** | | |
| 16–24 | 318 (50.1%) | 2,276 (39.4%) | | 1.82 (1.49,2.23);<0.001** | 1.51 (1.22,1.85);0.001** |
| 25–34 | 216 (34.0%) | 2,187 (37.8%) | | 1.29 (0.96,1.73);0.74 | 1.25 (0.99,1.59);0.06 |
| 35–49 | 101 (15.9%) | 1,318 (22.8%) | | Ref | ref |
| **Age (Mean, SD)** | 26.3 (8.0) | 28.3 (8.3) | | - - - | - - - |
| **Marital status[a]** | | | | | |
| Single/never married | 584 (92.0%) | 4,921 (85.1%) | <0.001** | 2.06 (1.61,2.63);0.27 | 1.35 (0.99,1.84);0.06 |
| Divorced/separated/widowed | 3 (<1%) | 26 (<1%) | | 2.00 (0.53,7.53);0.60 | 1.89 (0.51,7.01);0.31 |
| Married | 48 (7.6%) | 833 (14.4%) | | Ref | ref |
| **Education[b]** | | | 0.21 | | |
| Primary/lower | 115 (18.1%) | 910 (15.7%) | | 1.18(0.87,1.61); 0.26 | 1.35(0.98,1.86);0.06 |
| Secondary or higher | 520 (81.9%) | 4,868 (84.3%) | | Ref | ref |
| **Employment[c]** | | | <0.001** | | |
| Unemployed | 415 (65.5%) | 3,147 (54.5%) | | 1.58(1.24,2.01);0.001** | 1.34(1.06,1.69);0.02** |
| Employed | 219 (34.5%) | 2,623 (45.5%) | | Ref | ref |
| **Sex partners in past 12 months[d]** | | | <0.01** | | |
| 2 or more partners | 89 (14.1%) | 685 (11.9%) | | 0.87 (0.58,1.32);0.54 | 1.03 (0.65,1.64);0.88 |
| 1 partner | 333 (52.7%) | 3,665 (63.7%) | | 0.61 (0.38,0.97);0.05 | 0.70 (0.44,1.12);0.13 |
| No partners | 210 (33.2%) | 1,407 (24.4%) | | Ref | ref |
| **Alcohol use in past 3 months[e]** | | | 0.43 | | |
| Yes | 294 (46.4%) | 2,576 (44.6%) | | 1.08 (0.89,1.32);0.46 | 1.18 (0.99,1.39);0.06 |
| No | 340 (53.6%) | 3,204 (55.4%) | | Ref | ref |
| **Test Venue** | | | <0.01** | | |
| Mobile | 346 (54.5%) | 3,442 (59.5%) | | 1.76 (1.45,2.14);<0.001** | 1.88 (1.53,2.31);<0.001** |
| Home | 289 (45.5%) | 2,339 (40.5%) | | Ref | ref |

*n*, number of participants; SD, standard deviation

*Rao-Scott Chi-Square test;

**Significant at α = .05

[a]1 participant missing marital status.

[b]2 participants missing education level.

[c]12 participants missing employment status.

[d]27 participants missing number of sex partners.

[e]2 participants missing alcohol usage.

Additionally, some peri-urban communities also practice traditional circumcision. These efforts could explain the significant increase in baseline level of circumcision in the three years between BAIS and BCPP surveys. Social desirability bias in men's self-reported MC status is also plausible, as BCPP was widely presented as promoting MC.

Younger age, being unemployed, attaining secondary or higher education, and undergoing HIV testing at mobile venues rather than at home were associated with VMMC uptake in BCPP. Our findings were similar to other studies conducted in Botswana [25, 26]. VMMC programs have targeted adolescent and young boys with positive results [16, 24, 27, 28]. Mobile HIV testing venues were more likely to attract men while females were found at home [29].

Additional factors associated with VMMC uptake identified in other studies include religious affiliation and being in a serious relationship [25]. However, in our study, the magnitudes of the significant associations were low; the population circumcised through referral from HIV testing was fairly similar to the general participant population. This is a reassuring finding with respect to age, as exclusively reaching younger clients is neither difficult nor desirable for VMMC programs; older men have higher average HIV incidences and thus benefit more from circumcision, and were adequately reached through our approach. The association of greatest magnitude was with being reached through mobile venues, which is also potentially a reassuring finding as these are less labor-intensive and are already a more common practice than door-to-door testing [29, 30]. Finally, the association of uptake with being unemployed also suggests this approach could be a useful complement to the workplace-based VMMC mobilization approaches already recommended by WHO [31], reaching the men those approaches may miss.

The 10% uptake of MC services among uncircumcised men was lower than expected. Higher uptake may have been achieved if the intervention period had lasted longer than 6–8 weeks in each community. Men identified through HIV testing services who were offered a referral to MC services had a high refusal rate, and those accepting the referral had relatively low yield. However, 16% of men accepting the referral were circumcised during the study whereas less than 1% of men refusing referral were circumcised. A previous analysis from BCPP data comparing uptake of MC services in the intervention compared to the control arm of the study showed that the study MC activities substantially increased uptake over the study duration: the proportion of eligible men circumcised increased by 10% in the intervention arm compared to 2% in the standard of care arm (relative risk, 1.26; 95%CI:1.17,1.35) [32]. The data provided here substantiates these findings and offers insights into which study activities were beneficial. Given the simplicity of this intervention, a 16% yield of those accepting VMMC referrals from high-volume testing may be quite a substantial VMMC achievement.

The barrier to VMMC uptake most consistently identified in our study was the fear of pain, despite most men being aware of the benefits of VMMC. Pain is a known deterrent to MC, and the Ministry of Health is revisiting pain messaging in national MC promotional materials [12, 27]. While traditional or religious prohibitions or silence on MC may lead to lower circumcision prevalence, in our study, religious or traditional beliefs were not significant reasons for not being circumcised or not seeking VMMC [33, 34]. Other barriers identified in other studies include lack of knowledge of the benefits of circumcision, and need to abstain from sex for six weeks after the procedure [27, 34].

As a limitation, this study was conducted in rural and peri-urban settings and may not be fully generalizable to the general population. Also, the data were collected in 2013–2016. The findings remain relevant: Botswana missed the 2021 90% VMMC coverage target for 10–29 year olds, and there is no obvious reason why referral through HIV testing would be a less promising route to VMMC now than previously. However, it cannot be assumed that demographic associations have remained stable. Another potential limitation is potential incomplete or inaccurate documentation of mobilization efforts. Finally, after the brief surge in each community ended, some additional men may have been circumcised at local clinics as a result of study mobilization efforts; these circumcisions would not have been reported to the study, leading to underestimation of impact.

## Conclusion

Younger age, being unemployed, and undergoing HIV testing at mobile venues rather than at home were associated with current VMMC uptake. Short-term MC programs (6–8 weeks) can

increase coverage by up to 10%. HIV testing programs can increase the percent of men getting circumcised by offering information and referral to services for interested men. Given the simplicity and low cost of this intervention, the HIV testing programs appear to offer an effective platform for identifying uncircumcised men and offering information and encouragement to access services.

## Acknowledgments

We would like to thank the study participants, and VMMC and HIV testing and counseling teams who made the Botswana Combination Prevention Project possible.

**Disclaimer**: Preliminary results were presented at the International AIDS Society Conference (AIDS2018), Abstract #WEPEC235, Amsterdam, Netherlands, 23–27 July 2018.

## Author Contributions

**Conceptualization:** Etienne Kadima, Lisa A. Mills, Jan Moore, Pam Bachanas, Stephanie Davis, Conrad Ntsuape, Refeletswe Lebelonyane, Naomi Bock.

**Data curation:** Tafireyi Marukutira, Faith Ussery, Stephanie Davis.

**Formal analysis:** Faith Ussery, Lisa Block, Stephanie Davis.

**Investigation:** Tafireyi Marukutira, Faith Ussery, Etienne Kadima, Stephanie Davis, Tracey Schissler, Roselyn Mosha, Conrad Ntsuape, Naomi Bock.

**Methodology:** Faith Ussery, Lisa A. Mills, Pam Bachanas, Stephanie Davis, Refeletswe Lebelonyane, Naomi Bock.

**Project administration:** Tafireyi Marukutira, Etienne Kadima, Lisa A. Mills, Lisa Block, Stephanie Davis, Tracey Schissler, Roselyn Mosha, Onneile Komotere, Thebeyame Diswai, Refeletswe Lebelonyane, Naomi Bock.

**Supervision:** Tafireyi Marukutira, Faith Ussery, Etienne Kadima, Lisa A. Mills, Jan Moore, Lisa Block, Pam Bachanas, Stephanie Davis, Refeletswe Lebelonyane, Naomi Bock.

**Validation:** Tafireyi Marukutira, Faith Ussery, Stephanie Davis.

**Visualization:** Tafireyi Marukutira, Stephanie Davis.

**Writing – original draft:** Tafireyi Marukutira, Stephanie Davis, Naomi Bock.

**Writing – review & editing:** Tafireyi Marukutira, Faith Ussery, Etienne Kadima, Lisa A. Mills, Jan Moore, Lisa Block, Pam Bachanas, Stephanie Davis, Tracey Schissler, Roselyn Mosha, Onneile Komotere, Thebeyame Diswai, Conrad Ntsuape, Refeletswe Lebelonyane, Naomi Bock.

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
