## [Decision Letter · Decision Letter 0]

15 Dec 2021

PONE-D-21-21032

Male circumcision uptake during the Botswana Combination Prevention Project

PLOS ONE

Dear Dr. Marukutira,

Thank you for submitting your manuscript to PLOS ONE. After careful consideration, we feel that it has merit but does not fully meet PLOS ONE’s publication criteria as it currently stands. Therefore, we invite you to submit a revised version of the manuscript that addresses the points raised during the review process.

The manuscript has been evaluated by three reviewers, and their comments are available below. The reviewers carefully assessed the findings of this manuscript, and while they generally expressed interest in the findings of the study, they have raised a number of concerns. They feel the methodology should be expanded to include information regarding the measurement of the variables and provide more information about the statistical methods. Additionally, the reviewers also feel that the introduction should be further expanded to fully explain the rationale for the study.Of particular note, reviewer 2 raised concerns about the use of unique national identification numbers to identify and link individuals, and whether this could impact the confidentiality of patientsCould you please carefully revise the manuscript to address and respond to all of the comments raised?

We look forward to receiving your revised manuscript.

Kind regards,

Jamie Royle, PhD

Academic Editor

PLOS ONE

2. Please provide additional details regarding participant consent. In the ethics statement in the Methods and online submission information, please ensure that you have specified whether consent was written or verbal/oral. If consent was verbal/oral, please specify: 1) whether the ethics committee approved the verbal/oral consent procedure, 2) why written consent could not be obtained, and 3) how verbal/oral consent was recorded.

3. You indicated that you had ethical approval for your study. In your Methods section, please ensure you have also stated whether you obtained consent from parents or guardians of the minors included in the study or whether the research ethics committee or IRB specifically waived the need for their consent.

Furthermore,  please ensure you have included the registration number for the clinical trial referenced in the manuscript.

4. Thank you for stating the following in the Acknowledgments: Disclaimer Section of your manuscript:

“We would like to thank the study participants, and VMMC and HIV testing and counseling teams who made the Botswana Combination Prevention Project possible.

Disclaimer: This project has been supported by the President’s Emergency Plan for AIDS Relief (PEPFAR) through the Centers for Disease Control and Prevention (CDC) under the terms of Cooperative Agreements U2G GH000073 and U2G GH000419. The findings and conclusions in this report are those of the authors and do not necessarily represent the official position of the funding agencies. Preliminary results were presented at the International AIDS Society Conference (AIDS2018), Abstract #WEPEC235, Amsterdam, Netherlands, 23-27 July 2018.”

“This project has been supported by the President’s Emergency Plan for AIDS Relief (PEPFAR) through the Centers for Disease Control and Prevention (CDC) under the terms of Cooperative Agreements U2G GH000073 and U2G GH000419.”

“This project has been supported by the President’s Emergency Plan for AIDS Relief (PEPFAR) through the Centers for Disease Control and Prevention (CDC) under the terms of Cooperative Agreements U2G GH000073 and U2G GH000419.”

Reviewers' comments:

Reviewer's Responses to Questions

**Comments to the Author**

1. Is the manuscript technically sound, and do the data support the conclusions?

Reviewer #1: Yes

Reviewer #2: Partly

Reviewer #3: Yes

2. Has the statistical analysis been performed appropriately and rigorously? 

Reviewer #1: Yes

Reviewer #2: Yes

Reviewer #3: Yes

3. Have the authors made all data underlying the findings in their manuscript fully available?

Reviewer #1: Yes

Reviewer #2: Yes

Reviewer #3: No

4. Is the manuscript presented in an intelligible fashion and written in standard English?

Reviewer #1: Yes

Reviewer #2: Yes

Reviewer #3: Yes

5. Review Comments to the Author

Reviewer #1: General comments:

The study design and analytic methods are appropriate and performed at a high standard in this manuscript. The only comments I had were requests to add in a little more information on some of the methods used, especially adding in citations.

Specific comments:

1. (line 93) How were the 30 communities selected? Were these randomly selected from a wider pool of communities or were these purposely selected? Furthermore, how were the 15 selected from the 30?

2. (line 128) Please include the version and citation for Epi Info.

3. (line 133) Please provide a citation for the Rao-Scott Chi-Squared test.

4. (lines 136-137) Since there are many different ways to account for clustering in logistic regression models, I strongly suggest you provide just a little more information on the method used here. In SAS, this could be done with GENMOD, GLIMMIX, SURVEYLOGISTIC, and other procedures. The SAS documentation should have the proper methodological citation.

Reviewer #2: The authors use descriptive quantitative methods to describe male participants characteristics by baseline circumcision status and circumcision uptake among those uncircumcised at baseline, including among uncircumcised men referred from HIV testing services. Data are from a pair-matched community randomized HIV prevention trial to evaluate the effect of a combination prevention intervention on HIV incidence in Botswana. The analysis uses data from 15 intervention communities where VMMC data was collected. Analyzed data were collected between October 2013 and February 2016. Study findings indicate low rates of male circumcision uptake. Males who were younger, unemployed, or referred from a mobile HIV testing site were more likely to undergo circumcision. Reasons for declining circumcision included not being ready and fear of pain.

Major Issues

Introduction

1. A clear identification of the problem and strong rationale for the paper is not provided. For example, 22 million circumcisions in 10 years seems laudable. Why should we be concerned about slow uptake with what appears to be a high number of men circumcised?

2. In concert with identification of the research problem, a review of studies that have examined or addressed the problem would strengthen the paper.

3. Background information about male circumcision in Botswana (traditional and medical) and, if available, the study regions would also strengthen the introduction and help the reader better understand the study and cultural context. For example, what was the prevalence of male circumcision in Botswana at the time of the study? Was traditional circumcision being practiced? When was medical male circumcision for HIV prevention introduced in Botswana? Was there a target goal for proportion of males circumcised medically? Is Botswana one of the countries with slow uptake? If so, how does uptake in Botswana compare to the other priority countries?

4. How do the authors define “demand creation activities” and how do/can HIV testing services fit withing demand creation activities?

5. Regarding lines 73-75, rather than presenting demand creation activities in general, it would be helpful to have a brief description of each activity listed (interpersonal mobilization, mass media, targeted service delivery).

6. The purpose of the paper is not clearly stated. Is there a gap in knowledge that the authors seek to address? If so, what is the gap? Or is the intent to inform policy? Is it both?

Methods

1. Is the study a secondary data analysis or were the authors involved in BCPP? If the former, this should be stated.

2. Line 96-97: A limitation of the study is that the data are old, and findings may not be applicable in the current Botswana context.

3. Line 101: Unclear what is meant by “…follow-up for a 6-8 week period to cover the remaining 80% of the community residents.” Please clarify.

4. Lines 129-130: There is concern whether unique national identification numbers that were used to identify and link individuals across databases are available in the publicly available datasets. If they are, there is a serious risk of breach of confidentiality.

Results

1. The different ways males were informed about VMMC and whether they underwent circumcision is confusing. Please clarify.

Discussion

1. A key limitation of the data is that they are relative dated. It is likely that VMMC uptake has changed in important ways since the data were collected and therefore it is unclear that these data provide useful information for the current context. Some discussion around this limitation is needed.

2. Missing is a discussion concerning the meaning and importance of the key findings that younger age, being unemployed, and undergoing HIV testing at mobile venues were associated with VMMC uptake. Further, how do these findings relate to those of other studies.

3. Line 213: What is the prevalence of traditional circumcision vs medical circumcision in Botswana?

4. Line 217: What proportion of men were expected to be circumcised?

5. Earlier in the paper (see lines 94-95) the authors state that data on VMMC were only collected from the intervention communities. However, in lines 222-227, data on proportion of males circumcised in the standard of care arm are presented. Please clarify.

Reviewer #3: Male circumcision uptake during the Botswana Combination Prevention Project- PONE-D-21-21032

Abstract

This article’s main objective was to examine sociodemographic characteristics and referral procedures associated with VMMC uptake in the Botswana Combination Prevention Project (BCPP) and examined the effectiveness of referral of men to MC services from HIV testing venues. The article is more relevant to the setting of Botswana, where HIV/AIDS is still a concern. The abstract is well written and covers in a nutshell the key findings of the study.

Introduction

The authors have succinctly provided a background of the SMC programme in Botswana and have stated the gap which their study seeks to address. They have also provided the rationale for their study. Meanwhile my suggestion is that they should provide a more detailed explanation how their study differs with many previous studies on SMC that have been done in Botswana. What more information would the study add?

Methodology

Although the methodology of the study is well explained there are some few things to note. It is not clear how you measured the outcome and explanatory variables. That information is vital for making the reader to understand how variables were conceptualized and measured.

Results

Please kindly revise the interpretation of table 1. Table 1 presents the characteristics of men aged 16-49 already circumcised vs not circumcised at baseline, but the authors interpret the results of the table as if the results are for the multivariate models. These are just proportions and should not be interpreted using words such as more or less likely, to imply multivariate associations. As a result, the authors may need to revisit the interpretation of this table and use proper language. In the last sentence of this paragraph you refer to multivariate analysis, which table? Please consider ordering the results in a more logical way. Referring to multivariate when interpreting bivariate tables does not seem well.

Discussion

This section is written well but can be improved. The key finding of this study is that Younger age, being unemployed, and undergoing HIV testing at mobile venues rather than at home were associated with current VMMC uptake. This is not mentioned in the discussion section; my view is that the discussion should centre on the key findings of the study.

6. PLOS authors have the option to publish the peer review history of their article (what does this mean?). If published, this will include your full peer review and any attached files.

Reviewer #1: No

Reviewer #2: No

Reviewer #3: **Yes: **Mpho Keetile, Ph.D.

---

## [Author Response · Author response to Decision Letter 0]

23 Jan 2022

RESPONSE TO REVIEWERS

Response: Thank you and noted.

2. Please provide additional details regarding participant consent. In the ethics statement in the Methods and online submission information, please ensure that you have specified whether consent was written or verbal/oral. If consent was verbal/oral, please specify: 1) whether the ethics committee approved the verbal/oral consent procedure, 2) why written consent could not be obtained, and 3) how verbal/oral consent was recorded.

Response: Thank you. A written consent was required. This sentence was added in the ethics statement: “The VMMC procedure was conducted according to the standards for routine VMMC service delivery in Botswana and the WHO minimum package of services, including written informed consent, sexually transmitted infections screening and treatment, condom education and provision, and postoperative follow-up for documenting healing [11].”

3. You indicated that you had ethical approval for your study. In your Methods section, please ensure you have also stated whether you obtained consent from parents or guardians of the minors included in the study or whether the research ethics committee or IRB specifically waived the need for their consent.

---

## [Decision Letter · Decision Letter 1]

17 May 2022

Male circumcision uptake during the Botswana Combination Prevention Project

PONE-D-21-21032R1

Dear Dr. Marukutira,

We’re pleased to inform you that your manuscript has been judged scientifically suitable for publication and will be formally accepted for publication once it meets all outstanding technical requirements.

Kind regards,

Winnie K. Luseno, Ph.D.

Guest Editor

PLOS ONE

Additional Editor Comments (optional): To preserve transparency and uphold the integrity of the scientific process, I would like to acknowledge that I also served as Reviewer # 2 for your manuscript. In responding to all the reviewer's comments, I believe you have strengthened your paper.

Reviewers' comments:

Reviewer's Responses to Questions

**Comments to the Author**

1. If the authors have adequately addressed your comments raised in a previous round of review and you feel that this manuscript is now acceptable for publication, you may indicate that here to bypass the “Comments to the Author” section, enter your conflict of interest statement in the “Confidential to Editor” section, and submit your "Accept" recommendation.

Reviewer #1: All comments have been addressed

2. Is the manuscript technically sound, and do the data support the conclusions?

Reviewer #1: (No Response)

3. Has the statistical analysis been performed appropriately and rigorously? 

Reviewer #1: (No Response)

4. Have the authors made all data underlying the findings in their manuscript fully available?

Reviewer #1: (No Response)

5. Is the manuscript presented in an intelligible fashion and written in standard English?

Reviewer #1: (No Response)

6. Review Comments to the Author

Reviewer #1: (No Response)

7. PLOS authors have the option to publish the peer review history of their article (what does this mean?). If published, this will include your full peer review and any attached files.

Reviewer #1: No

---

## [Editor Report · Acceptance letter]

27 May 2022

PONE-D-21-21032R1 

Male circumcision uptake during the Botswana Combination Prevention Project 

Dear Dr. Marukutira:

I'm pleased to inform you that your manuscript has been deemed suitable for publication in PLOS ONE. Congratulations! Your manuscript is now with our production department. 

Kind regards, 

on behalf of

Dr. Winnie K. Luseno 

Guest Editor

PLOS ONE